# Phages Needed against Resistant Bacteria

**DOI:** 10.3390/v12070743

**Published:** 2020-07-10

**Authors:** Karin Moelling

**Affiliations:** 1Institute for Medical Microbiology, University Zurich, Gloriastr 30, CH-8006 Zurich, Switzerland; moelling@molgen.mpg.de; 2Max-Planck-Institute for Molecular Genetics, Ihnestr 63-73, D-14195 Berlin, Germany

**Keywords:** phage therapy, multidrug-resistant bacteria, case reports, regulation, probiotics

## Abstract

Phages have been known for more than 100 years. They have been applied to numerous infectious diseases and have proved to be effective in many cases. However, they have been neglected due to the era of antibiotics. With the increase of antibiotic-resistant microorganisms, we need additional therapies. Whether or not phages can fulfill this expectation needs to be verified and tested according to the state-of-the-art of international regulations. These regulations fail, however, with respect to GMP production of phages. Phages are biologicals, not chemical compounds, which cannot be produced under GMP regulations. This needs to be urgently changed to allow progress to determine how phages can enter routine clinical settings.

## 1. Introduction

Hospitals are becoming a site where one can catch multidrug-resistant bacteria. The number of patients dying from hospital infections due to antimicrobial resistance (AMR) is about 33,000 annually in Europe. Infections in Europe amount to 2.5 Mio, as described by the Robert Koch Institute, Berlin.

AMR arises because bacteria can change when treated with antibiotics, and resistance is developed to them. For the year 2050, the World Health Organization predicts that 10 million people will die of multidrug-resistant bacteria. New antibiotics are not a focus of pharmaceutical companies. We are facing a new global health crisis.

Without effective antimicrobials for the prevention and treatment of infections, many medical treatments become risky. These comprise organ transplantation, diabetes, major surgery such as caesarian sections or hip replacements, and chronic infections of organs such as the lungs and urinary tract. AMR bacteria are frequently in hospitals where people are treated with antibiotics, which lead to resistance.

An important influencing factor for the spread of antimicrobial resistance occurs by misuse on people and children. More recently, medical prescriptions are encouraged to reduce the free availability of antibiotics in many countries. Furthermore, it is not generally known that antibiotics are not effective against viral infections, which can lead to misuse. In addition, the prevention of diseases in animals by antibiotics is often applied to large animal batteries, comprising thousands of chickens or pigs. Their individual treatment is replaced by treating all animals. This is attracting attention and countermeasures. Moreover, the growth-promoting effect of antibiotics is forbidden in animals in many countries. However, antibiotics may lead to an increase in body mass and obesity in younger children who have undergone several cycles of antibiotics at an early age. 

One could start with phages against the most relevant bacteria for human use, published by World Health Organization (WHO) under the acronym ESKAPE, namely, *Enterococcus faecium*, *Staphylococcus aureus*, *Klebsiella pneumoniae*, *Acinetobacter baumannii*, *Pseudomonas aeruginosa*, and *Enterobacter*. These germs are considered to be the main causes of hospital infections Appendix A.

We need to study and use phages better, without the present restrictions, for the benefit of patients. Phages are friends, not foes.

A friend’s diabetic toes were cut off recently, and he was recommended by a hospital in Berlin to fly to Georgia for an alternative treatment. There, the Eliava Institute in Tiflis, Georgia, has existed since 1923, and it uses phages to cure resistant bacterial infections.

## 2. A Brief History of Phage Therapy 

For thousands of years, the river Ganges has been the source of a disease-curing activity, which may have been the basis for a religious ritual of the Hindus. The river was the sewage system of those days, containing bacteria and their viruses, the phages. If someone was infected with the bacteria present in the river, this may have led to a curing effect. The person who was sick could not catch the same disease again; however, if lucky, the person would have swallowed some water containing the viruses of the bacteria, the bacteriophages, or phages, which undergo a natural growth cycle. 

Lytic phages destroy the bacteria and are released into the water. Before the end of the 20th century, the British biologist Ernest Hanbury Hankin (1865–1939) had already described the curing effect of the water from the Ganges river by testing it against cholera bacterial cultures, which had been known from Robert Koch’s studies for some decades. The bacteria were lysed by a then-unknown activity that was present in the water. The bacteria did not grow in the unboiled water but continued growing when the water was boiled. This indicated to Hankin that there must have been some kind of activity that prevented bacterial growth in the normal Ganges river water that was thermosensitive, indicating some labile biological activity, which later turned out to be the phages [1].

All bacteria harbor phages, and within 24 h, about half of them can become lysed, destroy the bacteria, and release hundreds of new phages. If all bacteria disappear, the phages will die out.

It took 20 more years to understand what was going on. Two researchers, Frederick Twort (1877–1950) and Felix d’Hérelle (1873–1949), discovered, most likely independently, the agent that kills bacteria. D´Hérelle gave that activity the name bacteriophages or phages, assuming the agent was eating the bacteria, which was not true. The phages lysed the bacteria in order to replicate and be released from the bacteria for a new round of infection in a new bacterial host.

D’Hérelle continued to study this activity, and he applied the phages to numerous bacteria. He also noticed that he had to get the phages from local bacteria and that a cocktail of more than one phage was more likely to lead to success. He took the feces of a soldier in Paris and isolated the soluble fraction using a porcelain filter, the Chamberland filter, which kept the bacteria back. The soluble fraction contained the phages. He swallowed them to prove that they did not cause any harm and treated soldiers and sick children with bacterial diarrhea and cured them. D’Hérelle traveled to wherever infectious disease outbreaks occurred and treated patients with phages, namely, South America, Africa (Ruanda, Burundi, Congo), passengers on a French ship in the Suez Canal, Mexico, Africa, India (Assam), and Russia.

He went to Mexico and later to Argentina and isolated a bacterium *Coccobacillus* to kill locusts; he treated cholera outbreaks and pestis bacteria and disinfected water reservoirs by adding phage suspensions.

He published these results in his first paper on phages in 1917 [2]. 

Felix d’Hérelle was invited to Tbilisi, Georgia, in 1936 by the bacteriologist Georgi Eliava, his former student, who had initiated the foundation of what later became the Eliava Institute of Bacteriophages, Microbiology, and Virology in Tbilisi in 1923. Phages are being produced there to this very day. This was once a major business, and trials were performed with many participants, including controls. The enormous production scale, the science behind it, the quality of the trials, including controls, are, to some extent, underestimated today. Up to 1200 people were employed there at peak times and produced large amounts of phages. Methods were developed on how to store or ship the phages; band-aids and powder were also developed to cover wounds, and even pills were produced for easier transportation, up to 1.5 Mio pills per year. The military was a major recipient and driving force for large-scale productions and consumption. In 1939, in the Finnish–Russian war, 18,000 soldiers received phage mixtures, “cocktails” against anthrax, which were dripped into their open fractures and 80% of them recovered without amputations [3,4]. 

In Tbilisi, the use of phage therapy was continued because of the limitations of antibiotics and has been maintained until today. They combine phages with antibiotics that are available. The numerous studies performed for almost 100 years indicate that phages can be successful and have proven to be safe. Adverse events have never been reported.

## 3. Recent Phage Therapy Trials 

Till today, it is sewage water, preferentially from hospitals, that serves as the major source for finding new phages against bacteria. Where there are bacteria, there will also be the respective phages. Almost all bacteria are infected by phages, and they have a cycle of about a day—a peak of bacterial lysis will lead to a peak of phages and reduction of the bacteria, then the phages are reduced and can replicate again when the bacteria multiply [5]. 

There have been only very few clinical trials (Phase I, II, or III) of phages performed worldwide, according to the standards required by present regulatory authorities for approval of a drug. One of the exceptions was the treatment of chronic otitis by phages [6]. However, there are several individual case reports of “compassionate use”, which can be performed with the consent of the patient, doctors, and authorities if there are no other options in life-threatening diseases, according to the Helsinki Declaration (2013) [7]. 

The PhagoBurn trial was performed against large burns, supported by the European Commission, EU, within Framework 7. This trial was a combined European effort that included nine burn centers in European countries, designed in 2014. This effort deserves respect because it was initiated very early on, but difficulties occurred. Large burns are often infected with numerous bacterial types, so that the rule in their protocol, twelve kinds of phages for treatment of one target, could not be met. Furthermore, the GMP-production for the phages by the company Clean Cells Co. in Montaigu in France proved to be difficult and required most of the grant money [8].

Phages are preferentially applicable for open wounds, open fractures, and deep sores, such as gangrene. Thus, phage therapy has also been recently applied for the diabetic toes of 11 patients who were not in life-danger but had to face amputations and were saved from surgery [9]. The advocate and coauthor of this trial was Elizabeth (Betty) Kutter, who has been supporting phage progress and therapy for decades.

A soldier from Ukraine was treated in 2016 after an odyssey through several countries for two years. He had skin and bone defects and multiresistant *Pseudomonas aeruginosa* biofilms. He was treated in the military hospital in Berlin by the surgeon Ch. Willy, who applied several methods, including surgery, disinfectants, skin transplants, antibiotics, and a phage cocktail from Georgia, Pyo-Phage, directed against *St. aureus*, *Streptococcus*, *P. aeruginosa*, *E. coli*, and *Proteus*. The patient received three ampules per day. Finally, his drainage secretion was sterile, and he received a prosthesis [10].

Furthermore, the company Nestlé performed a clinical trial by using selected members of the well-known phage T4 against *Escherichia (E.) coli* bacteria for children suffering from diarrhea in Bangladesh. A clinical trial was performed in Bangladesh, which fulfilled all requests by the western legal authorities. However, the phage did not infect the bacteria of the sick children even though it had been pretested. The bacteria had changed during the preparation of the clinical trial requirements, and the phages did not match the bacteria any more [11]. 

Urinary tract infections (UTIs) were treated with phages in about 70 cases in Tbilisi in a collaboration between the Eliava Institute with a Swiss clinical investigator, Thomas Kessler, as described ([12]), with the results still to be reported. Recently, a proposal for a clinical trial built on this experience was applied for in Switzerland. UTI is not a deadly disease but amounts to about 40% of hospital-acquired infections. It allows the investigation of different testing parameters, regimens, development of resistance, determination of half-life of the phages and titers, which cannot be performed in life-threatening situations such as sepsis and is urgently needed for further studies. A new trial has been initiated in Switzerland.

One of the most spectacular cases that have been described is the phage therapy of a patient, Tom Patterson, with sepsis by multidrug-resistant *Acinetobacter baumannii*. He is a patient with one of the most famous and recent success stories, with an open-access public story on YouTube and broad media coverage. He was infected as a tourist in Egypt by the bacterium *Acinetobacter baumannii* and had health problems with pancreatitis, diabetes, and sepsis. After several weeks in a coma, he woke up after phages were injected locally and intravenously [13]. This was a wake-up call for phage therapy.

A similar problem with heart surgery was recently solved in Hannover, Germany. Phage therapy was applied to a patient with a history of heart surgery who developed sepsis two years later at the Clinic for Heart and Thorax Transplantation at the Medical University of Hannover (MHH). The phages were prepared inhouse and made endotoxin-free by Endo-Trap column chromatography and injected directly into the heart lesion. He recovered [14].

In a recent case, a 15-year-old girl with cystic fibrosis received a lung transplant in London, England, and antibiotics. Phage therapy was initiated against her multidrug-resistant bacterial infection, *Mycobacterium abscessus,* which had destroyed her lungs and still affected her skin and liver. She received three phage types; two of the phages were delivered from Graham Hatfull from Pittsburgh, who had organized a SEA–PHAGES project (Science Education Alliance–Phage Hunters Advancing Genomics and Evolutionary Science), where students accumulated and stored 15,000 phages (about 1800 of them were characterized). Two of them were bioengineered to make them lytic and able to kill the infected bacteria for the patient in London. This bacterium is distinct from *Mycobacterium tuberculosis*, the causative agent of tuberculosis (TB), which is, unfortunately, very difficult to treat with phages due to its encapsidation and is frequently antibiotic-resistant [15]. AMR is very frequent in people infected with tuberculosis, amounting to almost 500,000 cases globally. G. Hatfull wants to find a phage for the treatment of resistant-TB. 

A recent case was published in a Belgium newspaper, the “Saint-Luc baby”, a 13-month-old baby with liver and blood infections by multidrug-resistant bacteria. She received phages for 85 days by a military doctor, Colonel Patrick Soentjens, from the Military Neder-over-Heembeck Hospital near Brussels. The phages were described as “trained” and “tailor-made” and selected for this case. The production of the phages is worth mentioning because it was made as a “pharmaceutical compound” under well-defined conditions in a qualified pharmacy [16]. Such productions are needed in general.

## 4. Stool Transfer with or without Bacteria? 

For patients suffering from severe diarrhea and *Clostridium difficile* (recently reclassified as *Clostridoides difficile*) infection, a stool transfer, called fecal microbiota transplantation (FMT), can be performed with feces from a healthy donor. This is a well-known procedure, which we recently revitalized for patients suffering from untreatable diarrhea and multidrug-resistant microbiota in the gut [17,18]. Every year, 15,000 people die of refractory *C. difficile* infections in the USA. Only in one case, a recipient of FMT died (and another recovered) because the donor feces contained drug-resistant *E. coli* not tested for beforehand; however, this did not affect 22 other patients receiving feces from the same donor [19]. Testing of stool samples will reduce this risk in the future. FMT is used exclusively for *C. difficile* patients but not against other intestinal diseases and not against obesity. The method has become routine, saves lives, and should be made available for other intestinal disorders soon ([5,20]).

Bacteria are up to about 80% to 100% infected with phages, and the turnover of bacteria in the ocean occurs in about 24 to 48 h [21]. The microbiome in the oceans has been compared with that of the human gut microbiome, comprises about 10^12^ bacteria of about 1000 types, with a correspondingly estimated tenfold number of phages [17,21,22,23,24]. Furthermore, the composition and diversity of bacteria and phages differ in normal and nonhealthy human feces. 

More recently, a modified FMT was performed with phages only, without the bacteria from the donor stool for fecal transfer [25]. The results suggested a therapeutic efficacy of the bacteria-free stool preparation, likely due to phages against *C. difficile* infection, which needs to be verified with a larger cohort. One may expect industrial products to be admitted in the near future.

A novel question referring to the usefulness of phages comes from the knowledge of the role of the microbiome in the gut for the success of anticancer immunotherapy. It has been shown how important the microbiome is for therapeutic success against cancer; this also means the phages, which play a role. Certain bacteria are required for successful immunotherapy [26]. Phages and bacteria belong together. Mice depleted of their microbiome were used in some of these studies. This raises the question of the need for animal studies for studying phages or microbiomes. Could cancer patients be saved by treatment with FMT?

## 5. Regulations 

A severe drawback for evaluation of the efficiency of phages against diverse diseases is the strict requirements for GMP production for human patient use. The present rules by the European committee EMA (European Medicines Agency, former EMEA) or the country-linked authorities are too restrictive because they were developed for chemical compounds and are not adequate for biologicals. Phages may mutate during the procedure of production and treatment and may be replaced by other phages or combinations—no legal rules exist for such a medical compound.

Several groups have suggested and tested “compound” or “magistral” production in selected and authorized pharmacies [27]. This would be a great advantage because of less expense and local production. If all of the groups need GMP material, this would drastically limit progress. Presently, the individual compassionate trials are extremely complicated, expensive, and time-consuming, impossible for broader applications. The conditions need to be such that a sufficient number of trials can be performed to evaluate the potential success of phage therapies. The host range of phages for bacteria is limited, and cocktails are required to cope with an infection; sometimes, the type of phages need to be altered during a trial, which is not normally allowed within clinical protocols. We urgently need special rules for phage therapy.

Why is targeting the gut microbiota with replication-competent pre- or probiotics allowed? Why is the use of phages as food additives to improve antibiotic treatments not allowed once we know which ones to use? D’Hérelle swallowed a phage cocktail to convince his colleagues of the safety of phages before they treated children in Paris. Not a single adverse event caused by phages has ever been described. We only need to continue the research on the best cocktails for specific indications.

One may specify that “We need phage therapy now” [5] and ask for phages to be allowed as “dietary supplements” or to call them “phage supplements” or just probiotics, which contain phages anyway!

Phages are not part of a novel concept in virology as they have been known for more than 100 years. However, they need to be remembered and activated for investigations. Phages cannot be easily studied today because of the high regulatory requirements on GMP production. However, before we know how useful they may be, they have to be analyzed extensively. The history of phages, as described above, as well as the few individual case reports, indicate how many difficulties we may face before phages become routine. We need to urgently find out whether they can hold their promise.

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
