# Peer review of "Phages Needed against Resistant Bacteria"

_viruses, 2020, doi:10.3390/v12070743_

Round 1
Reviewer 1 Report
The author has argued for relaxing GMP production guidelines for phages. While it is true that phages are less understood mode of therapeutics, it is even more important to have good manufacturing practices for phages so they are reliably manufactured and studied in clinical settings. Already, there is a bias of publishing positive results and not the negative ones. On top of that, if we have to relax GMP standards, the clinical studies would produce only worthless data and eventually, this would hurt the field very badly as the prospect of phages becoming therapeutics against resistant bacteria would face a roadblock. This commentary is in that sense going against good science. The author has discussed some historic context of sewage, etc., which is also based on low scientific data and more on hypothesis. Also, the write up has issues with grammar and typos. Although this is a commentary, it does not seem to be based on proper logic and science. It contains too much personal opinion and less of the prospects of the field.
Author Response
I thank the reviewer. I apologize for the many mistakes which are corrected now by a program and the reading of a collague. I was pressed in time for submission. Sorry.
I added a statement in the Introduction (line 21- 23) perhaps this makes the urgency of this article clearer.
Numerous experiments are summarized and as clear as possible. The reader should unterstand the principles not the details. Therefore I did not change there anything.
I obviously have a different view on phages and on GMP than this reviewer. I added a sentence about the future as requested. I see the future in the Belgian approach, new insert line 228-229
The "proper logic" and "science" of this commentary is based on the past and some recent successes. I do not think important studies have been missed. The conclusion is , to use the Belgian approach
I am aware of the opinion of this reviewer shared by the majority of people and colleagues - but I would like to point out to a severe problem and point out to an upcoming serious global health crisis.
Reviewer 2 Report
This review manuscript describes history and current status of phage therapy. Considering the importance of phage therapy in the era of antibiotic resistance, it is timely and covers major aspect of phage therapy. Although the manuscript is well written, I would like to point the followings;
1. lines 11-12. I agree phages need to be produced under different regulations than current GMP, but there are GMP produced phages available. For example, Clean Cells in France and Armata in the US.
2. line 112. In PhagoBurn, the phage cocktail contained 12 different phages to treat P. aeruginosa infection, not one kind of phage.
3. There are numerous typos, eg. line 159, Mycobacterium, line 214, phage therapy, line 120, P. (need to be capitalized), line 18, 33,000, and many more.
Author Response
I thank the reviewer for his comments. I want to apologize for the mistakes I was pressed in time for submission. I used a correction program and the help of a colleague.
I corrected the number of phages in PhagoBurn line114 and inserted a sentence on GMP production by Clean Cells Co. line 115-116. . I was surprised that the reviewer wants this to be mentioned.Their GMP prodution consumed most of the gratn.
Reviewer 3 Report
Please provide me back the paper with a deep grammar correction of the English words (example line 29 replace "infectins" with "infections" and "diseses" with "diseases").
Author Response
I am sorry for all the mistakes I was pressed in time when I submitted. They should all be corrected by now. Thank you.
Round 2
Reviewer 3 Report
Line 21: substitute "antibiotics" with "them"
Line 21: add a comma after "2050" and replace "multidrug resistance" with "multidrug-resistance"
Line 25: delete the dot after "surgery"
Line 29: add a comma after "recently"
Line 30: add a comma after "furthermore"
Line 31: add a comma after "also" as in Line 34
Line 33: replace "There" with "Their"
Line 34: replace "growth promoting" with "growth-promoting"
Line 41: delete "them" after "use"
Line 42: add a comma before "not"
Line 43: is not clear the word "neighbour's" refers to
Line 56: add a hyphen between "then" and "unknown"
Line 62: replace "harbor" with "harbour"
Line 64: delete the comma after "understand"
Line 65: delete the comma after "1949)"
Line 70: delete the comma after "bacteria"
Line 71: replace "feces" with "faeces"
Line 74: replace "diarrhea" with "diarrhoea" and "traveled" with "travelled"
Line 81: replace the first "his" with "this"
Line 84: the phrase "Phages have been produced there to this very day" is not clear, please rewrite it
Line 91: add a comma after "18.000"
Line 94: add a comma after "Tbilisi"
Line 99: add "the" before "major"
Line 100: add a comma after "bacteria"
Line 101: add a comma before "and"
Line 104: replace "have" with "has" and "world-wide" with "worldwide"
Line 106: add "the" before "treatment"
Line 107: add "the" before "patient"
Line 111: it is not clear who or what "This" refers to
Line 120: replace "coauthor" with "co-author"
Line 127: add a comma after "Finally"
Line 130: replace "diarrhea" with "diarrhoea"
Line 131: add a comma after "Bangladesh"
Line 133: add "the" before "preparation"
Line 133: replace "clinical swiss" with "a Swiss clinical"
Line 138: delete the comma after "disease"
Line 139: add a hyphen between "hospital acquired"
Line 139: replace "testing different" with "different testing"
Line 141: there is space between "life -threatening", please delete it
Line 144: add a hyphen between "multidrug resistant"
Line 147: add a comma after "coma"
Line 150: delete "A" before "phage"
Line 155: add a comma after "case" and replace "15 year old" with "15-year-old"
Line 156: delete "A" before "phage"
Line 159: replace "organized" with "organised"
Line 161: replace "characterized" with "characterised"
Line 163: add commas before and after "unfortunately"
Line 166: correct the spelling of "Hatfull" and add "the" before "treatment"
Lines 175 and 178: replace "diarrhea" with "diarrhoea"
Line 177: replace "feces" with "faeces"
Line 178: replace "revitalized" with "revitalised"
Line 180: add a comma after "case" and delete the comma after ")"
Line 181: delete the dot after "E. coli"
Line 182: replace "feces" with "faeces"
Line 185: replace "diseases" with "disorders"
Line 186: add a comma before "and"
Line 191: replace "feces" with "faeces"
Line 192: add a comma after "recently"
Line 193: replace "fecal" with "faecal"
Line 199: delete "a" before "successful"
Line 201: replace "need of" with "need for"
Line 204: replace "serious" with "severe"
Line 207: delete the comma after "restrictive"
Line 211: replace "authorized" with "authorised"
Line 213: add a comma after "Presently"
Line 219: add a hyphen between "replication competent"
Line 223: add "the" before "cocktails"
Line 229: delete "the" before "high regulatory"
Line 231: replace "analyzed" with "analised"
Line 233: delete the comma after "out"
Author Response
thanks for the correction
The Manuscript has been revised accordingly.
This manuscript is a resubmission of an earlier submission. The following is a list of the peer review reports and author responses from that submission.